# Contrastive Natural Language Explanations for Multi-Objective Path Planning

**Raj Korpan,**[1*] **Susan L. Epstein**[2,3]

[1]Iona College
[2]The Graduate Center of The City University of New York
[3]Hunter College, The City University of New York
rkorpan@iona.edu, susan.epstein@hunter.cuny.edu

## Abstract

This paper introduces a flexible, scalable approach that generates contrastive explanations of navigation plans based on multiple objectives. These explanations in natural language describe a robot controller's beliefs, intentions, and confidence to any person who travels with or near the robot. A new multi-objective path planning algorithm generates optimal single-objective plans, evaluates each of them with respect to the other objectives, and selects one. The objectives that favored the selected plan over the others become reasons in the explanation. Extensive evaluation in simulation demonstrates the system's ability to produce diverse, readily understandable explanations that provide counterfactual examples.

## Introduction

Autonomous robots indoors must often plan paths that satisfy multiple competing objectives, such as speed and safety. To appear accountable and trustworthy to the people they encounter in the real world, these robots should offer clear but nuanced explanations of their intentions in natural language. Faced with multiple objectives, traditional path planners have either compromised among those objectives to select each step or relied on a mathematical combination of them hand-tuned for a particular environment. It is more difficult, however, to explain the full trajectory of such a plan to a human companion, especially one who is not an expert. A *contrastive* explanation provides an alternative counterfactual that may address a human questioner's concerns (Miller 2019). The thesis of this work is that humans' demonstrated preference for contrastive explanations should drive the planning process itself. This paper introduces a novel approach, where a set of single-objective planners each constructs an optimal plan and then votes to identify the plan that best suits them all. This enables our robot controller to advocate for its chosen plan in contrast to another through their underlying objectives. The resultant natural language explanations address the controller's beliefs, intentions, and confidence. We demonstrate this in a challenging real-world environment.

---

*The work reported here was performed while the first author was at The Graduate Center of The City University of New York. Copyright © 2021, Association for the Advancement of Artificial Intelligence (www.aaai.org). All rights reserved.

An agent's *mental model* captures the internal representations and thought processes of another agent in the same environment. Communication helps people construct a mental model of how the robot perceives and reasons about their shared space, and thereby helps to establish trust (Kulesza et al. 2013). Questions from a person about a robot navigator's plan arise from a gap in the human's mental model of the robot or from a mismatch in their beliefs (Shvo, Klassen, and McIlraith 2020). Production of explanations that reduce or eliminate this gap is a difficult problem because it is hard for a person to judge what knowledge the robot possesses, and difficult for a robot to convey its knowledge given the mismatch in its actuators and sensors (Thellman and Ziemke 2021). The contrastive explanations introduced here address a person's confusion about a robot navigator's plan, both to improve the person's trust in future plans and to build their understanding of the robot's reasoning mechanism.

A single-objective plan can be justified simply by a statement of that objective (e.g., "I decided to go this way because it is the shortest path"). Such an explanation, however, does not address the questioner's reason for asking, nor does it provide any spatial context. Here, a multi-objective path planner considers several reasons to construct a plan, and uses those objectives to provide a contrastive explanation (e.g., "Although I may come close to obstacles, I'd rather go this way because it is shorter"). Moreover, the system measures a selected plan's adherence to the objectives to describe the controller's confidence in it.

The next sections provide background and related work in multi-objective path planning and Explainable AI Planning (XAIP). Subsequent sections describes *VBMO*, our voting-based multi-objective path planning approach, and the contrastive plan explanation procedure. The final section presents empirical results and discusses future work.

## Background

An optimal graph-search algorithm finds the least cost path from the robot's current location to its target's location. Typically, the algorithm exploits a weighted graph that describes navigable two-dimensional space. Such a graph $G$ represents unobstructed locations there as vertices. An edge in $G$ between two vertices indicates that one can move directly between them, with a label for the cost to do so. For example, if the objective were to minimize path length, labels

could record the Euclidean distance between pairs of vertices. Without loss of generality, we cast optimization here as search for minimum cost.

A simple path planner $\mathcal{H}_\beta$ seeks a plan $P$ in $G$ that minimizes a single objective $\beta$, such as distance. A plan $P$ is *optimal* with respect to $\beta$ only if no other plan $P'$ has a lower total cost $\beta(P)$ for that objective, that is, for every other plan $P', \beta(P) \leq \beta(P')$. Even an industrial robot, however, is subject to error. Its sensors may report inaccurately due to lighting or reflective surfaces, and its motors may produce unintended movements, particularly where surfaces are imperfect. Both kinds of errors make it difficult to maneuver along a carefully chosen path. As a result, a path planner is often tasked with multiple objectives, such as distance and proximity to obstacles.

A *multi-objective* path planner $\mathcal{H}_B$ seeks a plan $P$ that performs well with respect to a set $B$ of objectives. If, for example, $B = \{\beta_1, \beta_2\}$, where $\beta_1$ is travel distance and $\beta_2$ is proximity to obstacles, $\mathcal{H}_B$ would seek a plan $P$ that scores well on both objectives. Because objectives may conflict, no single plan is likely to be optimal with respect to all of $B$. Typically, a potential plan will perform better with respect to some $\beta$'s and worse with respect to others.

Let $B = \{\beta_1, \beta_2, \ldots, \beta_J\}$ be a set of planning objectives with respective plan costs $\{\beta_1(P), \ldots, \beta_J(P)\}$ calculated in graphs labeled by their individual objectives. A plan $P_1$ *dominates* another plan $P_2$ ($P_1 \ll P_2$) when $\beta(P_1) \leq \beta(P_2)$ for every $\beta \in B$ and $\beta_j(P_1) < \beta_j(P_2)$ for at least one objective $\beta_j \in B$. Dominance is transitive, that is, if $P_1 \ll P_2$ and $P_2 \ll P_3$, then $P_1 \ll P_3$ (Pardalos, Migdalas, and Pitsoulis 2008). Among all possible plans, a non-dominated plan lies on the *Pareto frontier*, the set of all solutions that cannot be improved on one objective without a penalty to another objective (LaValle 2006). A typical multi-objective planner searches for plans that lie on the Pareto frontier and then an external decision maker chooses among them.

## Related Work

Transparent, intelligible communication enables a robot to gain social acceptance and reduce confusion about its abilities (Rosenfeld and Richardson 2019; Wallkötter et al. 2021). While previous work focused on communication with experts (Scalise, Rosenthal, and Srinivasa 2017), this work focuses on explanations for laypeople. Although explicable, understandable behavior (e.g., (Chakraborti et al. 2019; Huang et al. 2019)) is a topic of importance, it often comes at the cost of suboptimality. Instead, the robot controller described here produces plans that are both explainable and optimal with respect to at least one objective.

Recent XAIP approaches rely on classical planning (Grea, Matignon, and Aknine 2018; Krarup et al. 2019) or logic (Nguyen et al. 2020) to produce explanations. None of those, however, explains in natural language. Several approaches to sequential tasks explained the state-action-reward representation of Markov decision processes, but the resultant language was less human-friendly than our approach and was not directly based on human reasoning (Ramakrishnan and Shah 2016; Khan et al. 2011; Hayes and Shah 2017). Another approach used deep learning to produce natural expla-

nations for an autonomous vehicle, but required an annotated dataset for training and did not address indoor navigation (Kim et al. 2018).

A contrastive explanation compares the reason for a decision or plan against another plausible rationale (Hoffmann and Magazzeni 2019). Counterfactual reasons for behavior have been shown to improve trust and understanding (Lim, Dey, and Avrahami 2009). A recent human-subject study showed that people preferred explanations focused on the differences between the robot's planned route and their own expectations (e.g., "my route is shorter, but overlaps more and produces less reward") (Perelman, Evans III, and Schaefer 2020). Similar to our approach, other recent work provided contrastive explanations in natural language for multi-objective path planning modeled as a Markov decision process (Sukkerd, Simmons, and Garlan 2020). It considered fewer objectives, however, required a hand-labeled map, and was evaluated in much smaller environments.

An early approach to multi-objective optimization treated it as a single-objective problem for a simple weighted sum of the objectives (Zadeh 1963). Others addressed individual objectives in a weighted sum with constraints (Haimes 1973), minimum values (Lee et al. 1972), or ideal values (Wierzbicki 1980). The weighted sum approach has also been applied to the heuristic function of an optimal search algorithm (Refanidis and Vlahavas 2003). All this work, however, required a human expert with knowledge of the relative importance of the objectives to tune the weights (Marler and Arora 2010). Moreover, small changes in those weights can result in dramatically different plans.

Many have used metaheuristics (e.g., evolutionary algorithms) to find non-dominated solutions to multi-objective problems (Deb et al. 2002). Those approaches, however, do not guarantee optimality, require tuning many hyperparameters, and are computationally expensive (Talbi et al. 2012). Furthermore, as the number of objectives increases, the fraction of non-dominated solutions approaches one (Farina and Amato 2002) and the size of the Pareto frontier increases exponentially (Jaimes and Coello 2015). As a result, methods that seek Pareto dominance break down with more objectives because it becomes more computationally expensive to compare all the potential non-dominated solutions. VBMO avoids this computation on infinitely many points on the surface of the Pareto frontier. Instead, it only ever compares $|B|$ solutions because it transforms the multi-objective problem into a set of single-objective problems.

A*, the traditional optimal search algorithm, requires an admissible heuristic, one that consistently underestimates its objective (Hart, Nilsson, and Raphael 1968). Several approaches extend A* to address multi-objective search. Multi-objective A* tracks all the objectives simultaneously as it maintains a queue of search nodes to expand (Stewart and White III 1991). NAMOA* extends multi-objective A* with a queue of partial solution paths instead of search nodes, but it is slow, memory hungry, and does not scale well (Mandow and De La Cruz 2008). Multi-heuristic A* modifies A* to consider multiple heuristics, some of which can be inadmissible (Aine et al. 2016). It interleaves expansion of search nodes selected by an admissible heuristic with

**Algorithm 1:** VBMO planning algorithm

---
**Input:** *single-objective planners $J$, shared graph $G$*
**for** *each planner $j \in J$* **do**
    Set $j$'s graph $G_j$ to a copy of $G$
    Label edges in $G_j$ based on $j$'s objective
    Find optimal plan $P_j$ in $G_j$

**for** *each planner $j \in J$* **do**
    **for** *each planner $i \in J$* **do**
        $C_{ij} \leftarrow$ cost of plan $P_i$ in $G_j$
    Normalize plan scores $C_{ij}$ in [0,10]

**for** *each plan $P_i$* **do**
    $Score_i \leftarrow \sum_{j=1}^{J} C_{ij}$
$best \leftarrow argmin_i\ Score_i$
**return** $P_{best}$

---

expansion on search nodes selected by nonadmissible ones. This approach was extended to treat the expansion from nonadmissible heuristics as a multi-armed bandit problem (Phillips et al. 2015). Other work has addressed these issues of efficiency and scale but only for two objectives (Ulloa et al. 2020).

Other multi-objective approaches draw from social choice theory. For example, in multi-attribute utility theory a function evaluates the available choices and selects the one with greatest utility (Keeney, Raiffa, and Meyer 1993). The approach closest to ours formulated multi-objective path planning as a reinforcement learning problem, and voted to select among the actions available at a state based on the expected reward under each objective (Tozer, Mazzuchi, and Sarkani 2017). That approach, however, required hundreds of episodes of training and only considered an artificial $10 \times 20$ grid environment with four obstacles.

VBMO, the path planning algorithm introduced here, uses topologically identical graphs, each of which has a set of labels that represent a different objectives. VBMO constructs an optimal plan in each graph, evaluates each plan in every graph, and then selects the plan with lowest total cost across all of them. This avoids the limitations of other approaches because it addresses each objective independently and then evaluates the resultant plans from the perspective of each planner. The full trajectory of a VBMO plan is inherently explainable in natural language and readily provides contrastive reasons based on the planner's objectives, even in a finely-detailed graph for a large, obstacle-ridden environment. Given explanations of VBMO's plan, a non-expert human could discern the individual rationales that motivated it, and understand the way VBMO weighed these rationales to produce the plan, which would improve trust in future plans.

## Voting-based Multi-objective Path Planning

VBMO constructs multiple plans, each of which optimizes a single objective, and then uses range voting to select the plan that maximally satisfies the most objectives. Pseudocode for it appears in Algorithm 1. First, each single-objective plan-

Table 1: Scores $C_{ij}$ for six plans $P_i$ given six objectives $\beta_j$. Normalization ensures that each plan is optimal with respect to its own objective. VBMO selects the plan with minimum total $Score_i$, here $P_2$.

|       | $\beta_1$ | $\beta_2$ | $\beta_3$ | $\beta_4$ | $\beta_5$ | $\beta_6$ | $Score_i$ |
|-------|------|------|------|------|------|------|------|
| $P_1$ | 0.0  | 1.4  | 5.0  | 6.7  | 2.5  | 7.5  | 23.1 |
| $P_2$ | 1.2  | 0.0  | 1.2  | 1.1  | 10.0 | 1.1  | 14.6 |
| $P_3$ | 5.7  | 8.6  | 0.0  | 2.2  | 6.3  | 5.0  | 27.8 |
| $P_4$ | 10.0 | 7.1  | 10.0 | 0.0  | 3.8  | 10.0 | 40.9 |
| $P_5$ | 5.7  | 10.0 | 2.0  | 10.0 | 0.0  | 6.3  | 34.0 |
| $P_6$ | 2.9  | 10.0 | 2.0  | 1.1  | 10.0 | 0.0  | 26.0 |

ner modifies a copy of the shared graph to reflect its objective in the edge weights. Then VBMO constructs an optimal plan $P$ in that modified graph. In this way, each submitted plan is guaranteed to be optimal for at least one objective.

Once it assembles the set of submitted plans $\mathcal{P}$, VBMO uses each planner's objective to evaluate all of them. Because each planner's underlying graph has the same topological structure (vertices and edges), every vertex $v$ in any plan is known to all the planners. To evaluate planner $\mathcal{H}_i$'s plan $P_i = \langle v_1, v_2, \dots, v_m \rangle$ from the perspective of planner $\mathcal{H}_j$ with objective $\beta_j$, VBMO sums the edge costs from the same sequence of vertices in $\mathcal{H}_j$'s own graph. In this way, each planner $\mathcal{H}_j$ uses its own objective to calculate a *score* $C_{ij}$ for each stored plan $P_i$.

To avoid any biases that would be introduced by the magnitude of an objective's values, all scores from any $\mathcal{H}_j$ are normalized in $[0, 10]$. Because VBMO seeks to minimize its objectives, a score $C_{ij}$ near 0 indicates that plan $P_i$ closely conforms to objective $\beta_j$, while a score near 10 indicates that $P_i$ strongly opposes $\beta_j$. Once every planner scores every plan, the plan $P_{best}$ with the lowest total score from all $J$ planners is selected by range voting:

$$P_{best} = \underset{P_i \in \mathcal{P}}{argmin} \sum_{j=1}^{J} C_{ij} \qquad (1)$$

Ties are broken at random. An example appears in Table 1.

**Theorem.** *Algorithm 1 constructs at least one plan guaranteed to be on the Pareto frontier.*

*Proof* by induction on the number of objectives $J$:
Consider first $J = 2$ with objectives $B = \{\beta_1, \beta_2\}$ and respective plans $P_1$ and $P_2$. By definition, planner $\mathcal{H}_j$'s plan $P_j$ optimally minimizes its objective, that is, $\beta_j(P_j) \leq \beta_j(P_k)$ for every $P_k \in \mathcal{P}$. Another planner $\mathcal{H}_k$ can score equally well on $\beta_j$, but cannot score lower than $\beta_j(P_j)$; otherwise, search would have returned $\mathcal{H}_k$'s plan to $\mathcal{H}_j$. Thus, there are only four possible cases

- *Case 1:* $\beta_1(P_1) = \beta_1(P_2)$ and $\beta_2(P_2) = \beta_2(P_1)$
- *Case 2:* $\beta_1(P_1) < \beta_1(P_2)$ and $\beta_2(P_2) = \beta_2(P_1)$
- *Case 3:* $\beta_1(P_1) = \beta_1(P_2)$ and $\beta_2(P_2) < \beta_2(P_1)$
- *Case 4:* $\beta_1(P_1) < \beta_1(P_2)$ and $\beta_2(P_2) < \beta_2(P_1)$

In case 1, both plans are non-dominated; $P_1 \not\ll P_2$ and $P_2 \not\ll P_1$ because neither's score is strictly less than the other on any objective. In case 2, $P_1 \ll P_2$ and $P_2 \not\ll P_1$, so $P_1$ is non-dominated. In case 3, $P_1 \not\ll P_2$ but $P_2 \ll P_1$, so $P_2$ is non-dominated. Finally, in case 4, $P_1 \not\ll P_2$ and $P_2 \not\ll P_1$, so both plans are non-dominated. Thus, when $J = 2$ there is always at least one plan that is non-dominated (i.e., on the Pareto frontier).

Assume now that for $J = k$ objectives $B = \{\beta_1, \ldots, \beta_k\}$ with optimal plans $\{P_1, \ldots, P_k\}$, one of them, plan $P_n$, is non-dominated. Then, by definition, there is no other plan $P_i$ such that $\beta(P_i) \leq \beta(P_n)$ for all $\beta \in B$ and for some $j \neq k, \beta_j(P_i) < \beta_j(P_n)$. Consider now $J = k + 1$, where we introduce one additional objective $\beta_{k+1}$ and its optimal plan $P_{k+1}$. With respect to dominance, there are three possible relationships between $P_n$ and $P_{k+1}$. If $P_n \ll P_{k+1}$, then $P_n$ remains on the Pareto frontier because it is still non-dominated. If $P_{k+1} \ll P_n$, then $P_{k+1}$ is on the Pareto frontier because transitivity ensures that it is not dominated by any other plan. Finally, if $P_n \not\ll P_{k+1}$ and $P_{k+1} \not\ll P_n$, both plans are non-dominated. Hence, at least one of $P_n$ or $P_{k+1}$ is non-dominated and lies on the Pareto frontier. ∎

In summary, VBMO is an efficient multi-objective path planning approach that always identifies and then selects a plan on the Pareto frontier, without reliance on finely-tuned weights. Theorem 1 proves that, given a set of dominated and non-dominated plans, VBMO voting will always select a non-dominated plan because non-dominated plans score lower with respect to at least one objective and therefore have a lower total score. Given $J$ objectives, VBMO has complexity $\mathcal{O}(J^2)$ because each plan is evaluated under each objective.

## Contrastive Plan Explanations

It has long been argued that instead of building systems to explain black-box models, models should deliberately be built to be interpretable (Rudin 2019), especially in a robotic context (Arnold, Kasenberg, and Scheutz 2021). VBMO's planning procedure makes it an inherently interpretable system without the need for any additional computation. It generates a set of single-objective plans and their scores with respect to all the objectives under consideration. These plans support readily constructed counterexamples for comparison to produce contrastive explanations.

Given objectives $B = \{\beta_1, \ldots, \beta_J\}$ with associated plans $\mathcal{P} = \{P_1, \ldots, P_J\}$, our explanation generator considers the relative adherence of any plan to those objectives. Each objective $\beta_k$ is associated with its own partition of [0,10] into bins that reflect how closely any plan diverges from it. For example, if the partition for $\beta_k$ were $\{[0, 2), [2, 7), [7, 10]\}$, then $\beta_k(P_i) = 0$ would place $P_i$ in the first bin and $\beta_k(P_j) = 4.7$ would place $P_j$ in the second. The generator associates these bins with natural language that describes the adherence to $\beta_k$. To do so it uses the function $\mathcal{L}$, which maps a score to a partition and outputs the language associated with that bin. In our example, if $\beta_k$'s three bins were translated as "a lot," "somewhat," and "a little," then $\mathcal{L}(\beta_k(P_i)) =$ "a lot" and $\mathcal{L}(\beta_k(P_j)) =$ "somewhat."

To compare two plans $P_i$ and $P_j$, the generator partitions their scores $C_{ik}$ and $C_{jk}$ from each $\beta_k$, and bins the scores for the two plans based on each associated partition. It then identifies those objectives $\beta_\ell$ where $\beta_\ell(P_i) < \beta_\ell(P_j)$, and those $\beta_g$ where $\beta_g(P_i) > \beta_g(P_j)$. Recall that VBMO minimizes, so the $\beta_\ell$'s are objectives that *favor* $P_i$ and the $\beta_g$'s favor $P_j$. This method excludes any objective $\beta_k$ for which $\beta_k(P_i) = \beta_k(P_j)$ because VBMO focuses on how plans differ. Finally, the generator produces a contrastive explanation for a selected plan with natural language descriptions $\mathcal{N}(\beta)$ of the objectives that favor or oppose the plan and the language from $\mathcal{L}$ that associates its scores with those objectives. To do so it instantiates this template:

Although another way may be $[\mathcal{L}(\beta(P_j))\ \mathcal{N}(\beta)]$,

I believe my way is $\{\mathcal{L}(\beta(P_i))\ \mathcal{N}(\beta)\}$.

With appropriate punctuation and conjunctions, the portion in square brackets is repeated for every $\beta \in B$ where $\beta(P_i) > \beta(P_k)$, and the portion in curly brackets for every $\beta \in B$ where $\beta(P_i) < \beta(P_k)$. For example, "Although another way may be somewhat shorter and a little less crowded, I believe my way is a lot safer, somewhat less obstructed, and a little more familiar." The order of the $\beta$'s is randomized to encourage unique explanations. The first line in the template is omitted if every objective prefers $P_i$.

While the contrastive explanation above describes the robot controller's beliefs and intentions, it does not address its confidence in $P_i$, the plan it selected from all the plans in $\mathcal{P}$. To do so, the generator uses two metrics: overall preference for the plan and an overall adherence to the objectives. Overall preference for a chosen plan compared to the others is defined as a $t$-statistic across all total scores. Recall that $P_i$'s total score is $Score_i = \sum_{j=1}^{J} C_{ij}$. Let $\mu_{\mathcal{C}}$ be the average total score for all plans in $\mathcal{P}$ and $\sigma_{\mathcal{C}}$ be their standard deviation. The *overall preference* for the selected plan $P_i$ is

$$\tau_i = \frac{Score_i - \mu_{\mathcal{C}}}{\sigma_{\mathcal{C}}} \qquad (2)$$

Overall preference $\tau_i$ for plan $P_i$ indicates how much more the objectives as a group prefer it to the other plans in $\mathcal{P}$. Because VBMO minimizes total score, ideally $\tau_i$ is negative and has a large absolute value, to indicate that it is far below the mean total score. In Table 1, for example, $\mu_{\mathcal{C}} = 27.7$ and $\sigma_{\mathcal{C}} = 9.05$, so the overall preference $\tau_2$ for $P_2$ is $-1.45$. This indicates a relatively strong preference for $P_2$ over the other plans because its total score is more than one standard deviation from the mean.

A plan's adherence to the objectives in $B$ can be measured several ways. Intuitively, it should include how often the plan meets the objectives well and poorly. A simple approach would count $W_i$, how many objectives the plan does well on (e.g., $C_{ij} < 1$) and $X_i$, how many it does not do well on (e.g., $C_{ij} > 9$). $W_i > X_i$ would then indicate a high adherence, $W_i = X_i$ a medium adherence, and $W_i < X_i$ a low adherence. This approach removes the nuances of VBMO's range voting; it replaces sums of real-valued scores from the objectives with combined approval voting, where a voter can express only approval, disapproval, or indifference (Felsenthal 1989). In Table 1, for example, it would find $W_2 = 1$ and $X_2 = 1$ for $P_2$. That indicates a medium adherence to

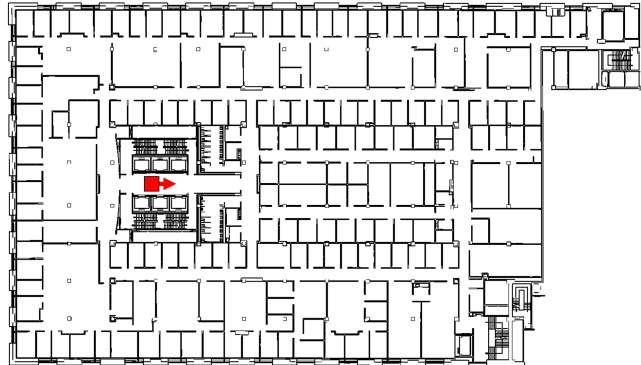

Figure 1: Floor plan of a large office environment with the robot's initial pose in red, next to the elevators that enter the floor.

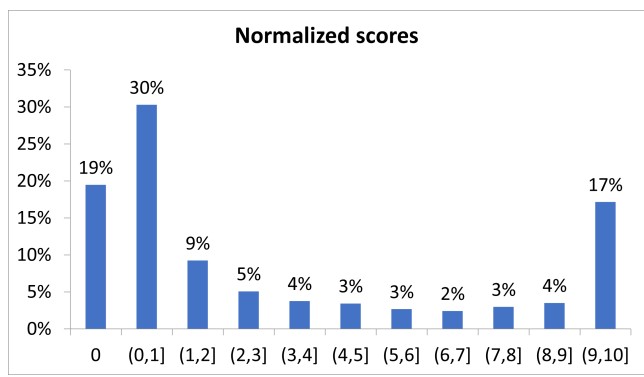

Figure 2: Distribution of normalized plan scores $C_{ij}$ across all the planners.

the objectives, even though in the range [0,10], 5 out of 6 objectives scored $P_2$ near 1. Another approach could transform VBMO's scores into a ranking and then use a measure of group coherence (Gehrlein and Lepelley 2016) but that too would ignore the real values.

Instead, the approach used here adapts a measure of inter-rater agreement. Typically, inter-rater agreement is measured with the $\kappa$ statistic, which compares ratings from two raters on a set of items. Here, we treat each objective $\beta$ as a rater that evaluates each plan and produces a score (i.e., its rating). While $\kappa$ indicates whether two objectives score plans similarly, it is not a good measure of the agreement among all the objectives on a single plan. Another measure is the average deviation index, which compares the ratings from multiple raters on a single item (Burke and Dunlap 2002). This works well as a metric of adherence because it accounts for the continuous values of VBMO's scores. Let $M_i$ be the median of the scores $C_{ij}$ for plan $P_i$ from all objectives in $B$. The adherence $\alpha_i$ of plan $P_i$ is

$$\alpha_i = \frac{\sum_{j=1}^{J} |C_{ij} - M_i|}{J} \qquad (3)$$

Smaller values of $\alpha_i$ indicate strong agreement among the objectives with respect to a plan. A ceiling of $0.2max(C_{ij})$ has been proposed to indicate agreement (Burke and Dunlap 2002). In Table 1, $M_2 = 1.15$, so $P_2$'s adherence $\alpha_2$ is 1.70. This is below the ceiling of 2.0, and so indicates strong agreement among the objectives in favor of the selected plan.

To bin their values for plan $P_i$ and assign natural language with $\mathcal{L}$, we associate both $\tau_i$ and $\alpha_i$ with partitions of the real numbers. For example, $\tau$ can be partitioned as $\{(-\infty, -0.75], (-0.75, -0.5], (-0.5, +\infty)\}$, with associated language "really," "somewhat," and "not really," and $\alpha$ with $\{[0, 1.5], (1.5, 2.5], (2.5, +\infty)\}$, with associated language "certain," "somewhat certain," and "conflicted." Instead of a ceiling that partitions $\alpha$ into two bins, we use three bins to more finely distinguish the adherence values. Each bin's associated language is classified by its *sentiment*, the affective state of the text, as positive or negative (Kim and Hovy 2004)). For example, "really," "somewhat," "cer-

tain," and "somewhat certain" are positive and "not really" and "conflicted" are negative.

If the sentiments expressed by $\mathcal{L}(\tau_i)$ and $\mathcal{L}(\alpha_i)$ agree on $P_i$, the description of the robot controller's confidence in its plan instantiates this template:

I'm $\mathcal{L}(\tau_i)$ sure I want to follow this plan,

and I'm also $\mathcal{L}(\alpha_i)$ about my reasons to go this way.

In Table 1's example, the two measures share the same sentiment for $P_2$ so the confidence explanation would be "I'm really sure I want to follow this plan, and I'm also certain about my reasons to go this way." Otherwise, if the sentiments do not agree, the description instantiates this template:

Although I'm $\mathcal{L}(\tau_i)$ sure I want to follow this plan,

I'm $\mathcal{L}(\alpha_i)$ about my reasons to go this way.

For example, the measures in Table 1 disagree about $P_1$. It has the second lowest total score, with $\tau_1 = -0.51$ and $\alpha_1 = 2.55$ so the explanation for $P_1$ would be "Although I'm somewhat sure I want to follow this plan, I'm conflicted about my reasons to go this way."

## Empirical Results

VBMO and its contrastive explanations have been evaluated in extensive simulation experiments with an autonomous robot controller for navigation in large, complex, indoor environments (Epstein and Korpan 2019). Written for ROS, the robot operating system, the simulator places our industrial robot on the fifth floor of a real-world office building. This environment is the size of a Manhattan city block ($110 \times 70m$) with 180 rooms of various sizes and several intersecting hallways. The controller overlays a fine grid on the environment's architectural floor plan (shown in Figure 1) to create VBMO's shared underlying graph.

Our controller also learns a cognitively-based spatial model while it travels, much the way a human would (Ishikawa 2021). Table 2 shows its eight planning objectives $B$. The first four are based on commonsense principles; the others reference the learned cognitively-based spatial model with language appropriately drawn from its constructs (Taylor and Tversky 1996; Talmy 2007). Given a target and planning objectives $B$, the controller uses VBMO to modify $|B|$ copies of the shared graph and constructs the set of plans

Table 2: VBMO's planners and their objectives

| Planner | Objective | Language $\mathcal{N}$ |
|---|---|---|
| FAST | Minimize distance traveled | "shorter" |
| SAFE | Avoid travel near obstructions | "less obstructed" |
| EXPLORE | Avoid travel along previous paths | "less familiar" |
| NOVEL | Avoid areas covered by the learned model | "more likely to reveal new areas" |
| TRAFFIC | Focus on small frequently-traveled areas | "more well-traveled" |
| HALLWAY | Exploit frequently-traveled vertical, horizontal, and diagonal routes | "more aligned with previous routes" |
| CIRCLE | Exploit a model of unobstructed circular areas | "more open" |
| TRACE | Follow refined versions of previous paths | "closer to ways we've gone before" |

Table 3: Average normalized plan scores $C_{ij}$ for each planner in every graph. Each plan is optimal in its own graph (i.e., $C_{ii} = 0.0$). The last column is the frequency with which a planner was selected by VBMO in 200 tasks.

| | $\beta_{Fast}$ | $\beta_{Safe}$ | $\beta_{Explore}$ | $\beta_{Novel}$ | $\beta_{Traffic}$ | $\beta_{Hallway}$ | $\beta_{Circle}$ | $\beta_{Trace}$ | Frequency |
|---|---|---|---|---|---|---|---|---|---|
| FAST | 0.00 | 0.20 | 3.53 | 5.30 | 1.74 | 1.46 | 1.30 | 1.93 | 32.4% |
| SAFE | 0.00 | 0.00 | 3.60 | 5.52 | 1.46 | 1.44 | 1.18 | 1.55 | 27.5% |
| EXPLORE | 3.42 | 3.62 | 0.00 | 4.92 | 4.94 | 4.63 | 3.89 | 4.52 | 4.9% |
| NOVEL | 8.07 | 8.27 | 5.47 | 0.00 | 9.08 | 8.38 | 9.44 | 9.38 | 0.0% |
| TRAFFIC | 1.47 | 1.64 | 5.60 | 7.06 | 0.00 | 2.26 | 0.91 | 1.02 | 16.5% |
| HALLWAY | 1.74 | 1.91 | 5.86 | 6.78 | 1.88 | 0.00 | 1.46 | 1.89 | 6.6% |
| CIRCLE | 1.83 | 2.00 | 5.45 | 8.60 | 0.80 | 2.34 | 0.00 | 0.90 | 0.5% |
| TRACE | 1.88 | 2.05 | 5.98 | 7.68 | 0.41 | 2.32 | 0.76 | 0.00 | 11.5% |

$\mathcal{P}$. It then evaluates each plan with respect to every objective and selects a plan. Finally, our controller generates an explanation for each plan.

At the start of an experiment, the robot controller receives a sequence of 40 randomly-selected target locations. For the first target, the robot's initial *pose* (location and orientation) faces east between the elevators. For all other targets, its initial pose is its final pose on the previous target. Table 3 reports the scores $C_{ij}$ for each planner's plan in the others' graphs, averaged over five different sets of 40 targets (200 tasks in all). Some planners perform well in another's graph (e.g., FAST and SAFE score each other's plans near 0). Only NOVEL scores poorly (i.e., above 9) fairly often. This is because its objective seeks to explore areas of the environment not captured in the learned spatial model, whereas four of the planners exploit that model. As a result, FAST and SAFE were most often selected by VBMO, and NOVEL never was.

The bimodal distribution in Figure 2 describes the overall distribution of all 12800 (8 objectives × 8 plans × 200 tasks) normalized plan scores $C_{ij}$ for all planners. It clearly indicates that most pairs of plans with distinct objectives either strongly conform to an objective or strongly oppose it. This suggests that the controller's objectives are sufficiently different to produce a diverse set of plans. Nearly 7% of the time, one objective scored another objective's plan as 0, which occurs when two plans are identical. Figure 3 is the distribution of the total scores $Score_i$ for the 1600 (8 × 200) generated plans. A total score of 0 would represent optimal adherence to all the objectives, but most often the selected plan's total score lies in (10, 20] which indicates some opposition to it.

Generation of contrastive explanations requires pairwise plan comparisons. Figure 4(a) reports on 11200 (200×8×7)

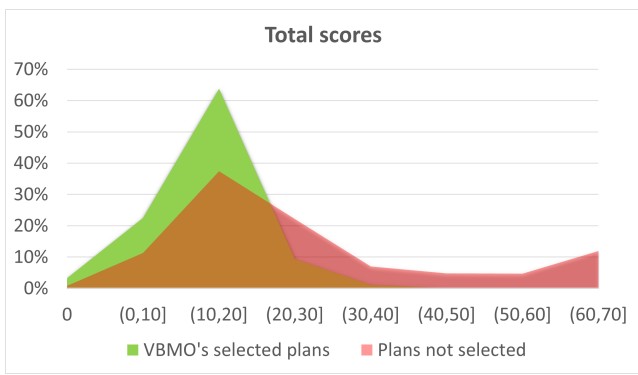

Figure 3: Distribution of total score $Score_i$ for plans $P_i$ in VBMO. Because $0 \leq C_{ij} \leq 10$ and VBMO scores each plan in seven other graphs, $Score_i \leq 70$. VBMO's selected plans have a better (i.e., lower) average $Score_i$ (13.8) than the unselected plans (27.0).

pairs of distinct objectives $\beta_i$ and $\beta_j$. It shows how often, when plan $P_i$ is compared to plan $P_j$, the objectives favor $P_i$, favor $P_j$, or have no preference between the two. The results show that a majority of the objectives often favor FAST's and SAFE's plans over other plans, that a majority oppose plans from EXPLORE and NOVEL, and that they split about evenly on plans from the other planners.

Figure 4(b) examines how often an objective favors one of the 200 selected plans over the 7 alternatives to it. In those 1400 tests, on average 4.8 objectives favored VBMO's selected plan and 2.3 favored an alternative. When each of the 1400 rejected plans is compared to the 7 others, fewer objectives favor the rejected plan and more objectives favor the

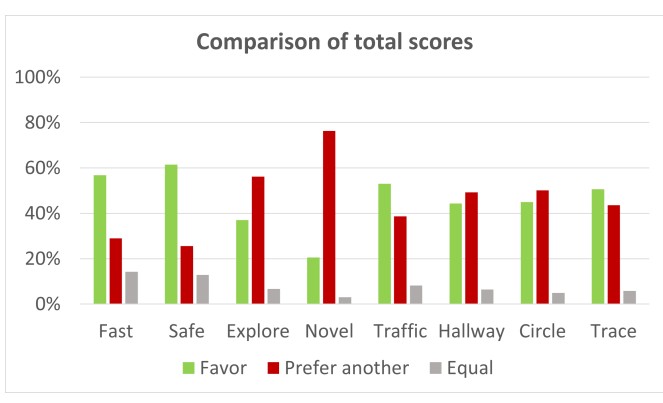

(a) Each planner's support from the other planners

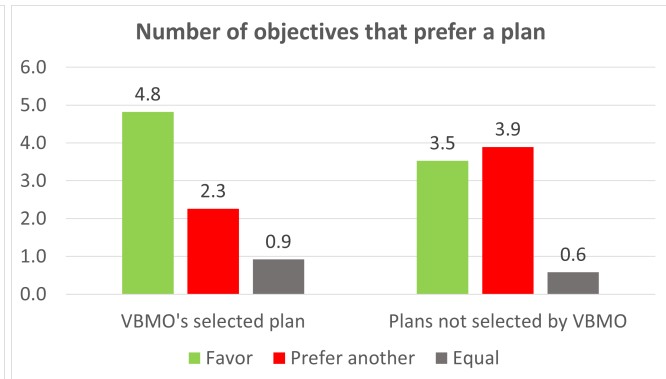

(b) Support for selected and rejected plans from the other planners

Figure 4: How a specific plan $P$ was viewed by other planners' objectives on 200 tasks. Objectives favor $P$ (green), favor some other plan (red), or have no favorite (gray). (a) From the perspective of each planner, how a plan it had submitted was scored by the other planners. For example, when a plan from EXPLORE was evaluated by each of the 7 other objectives, 37% of them favored $P_{Explore}$ and 56% favored an alternative. (b) On average, a majority of the objectives preferred the selected plan and about half the objectives often opposed a plan that was not selected.

alternative plan.

For evaluation, the system generated contrastive explanations for every pair of plans. As in the earlier examples, value partitions were $\{[0, 2), [2, 7), [7, 10]\}$ for all objectives $\beta$ with associated language "a lot," "somewhat," and "a little"; $\{(-\infty, -0.75], (-0.75, -0.5], (-0.5, +\infty)\}$ for overall preference $\tau_i$ with associated language "really," "somewhat," and "not really"; and $\{[0, 1.5], (1.5, 2.5], (2.5, +\infty)\}$ for adherence $\alpha_i$ with associated language "certain," "somewhat certain," and "conflicted." 96.8% of the explanations were unique, behavior consistent with the $Score_i$ distributions and the data on objectives' preferences. Explanations averaged 51.2 words and reading grade level averaged 7.8 on the Coleman-Liau index (CLI) (Coleman and Liau 1975), which indicates that they should be understandable to a layperson. One generated explanation was "Although another way may be a lot more well-traveled, a lot closer to ways we've gone before, somewhat more aligned with previous routes, and somewhat less familiar, I believe my way is a little more likely to reveal new areas, a lot more open, a lot shorter, and a lot less obstructed."

We also examined the values of the two confidence metrics, overall preference $\tau_i$ and adherence $\alpha_i$. Figure 5 shows the distribution of $\tau_i$ values for each plan $P_i$. VBMO's selected plan $P$ has an average overall preference of $-0.7$, which indicates that it is often "really" or "somewhat" preferred. For plans not selected by VBMO, this distribution peaks at about $-0.4$ with a right skew that indicates many other much less supported plans. For each plan $P_i$, whether it was selected or rejected, 35% of the values for $\alpha_i$ fall in $\{[0, 1.5]$ and nearly 45% of plans fall in $(1.5, 2.5]$. Thus, the majority of plans are reasonably certain.

With $\tau$ and $\alpha$ each partitioned into three bins, the confidence explanation template can produce only nine different explanations. All nine were generated during the five runs of the experiment. In 152 instances (19 tasks $\times$ 8 plans) con-

fidence explanations were not generated because all plans had equal total scores. For the 1448 other plans, confidence explanations averaged 20.6 words with a 5.6 reading grade level. Table 4 shows that 8.2% of all plans had overall preference $\mathcal{L}(\tau_i) =$ "really," to be expected because this data includes seven times more rejected than accepted plans. The selected plans, however, were "really" preferred 29.4%. Moreover, no selected plan was ever described as both "not really" and "conflicted." Across all plans, the confidence explanation most often produced was "I'm not really sure I want to follow this plan, and I'm only somewhat certain about my reasons to go this way."

## Discussion

VBMO's current explanations address two important questions: "Why did you select that plan?" and "How confident are you about your plan?" The same approach could be easily extended to address other important questions from a human companion, such as "Why don't we go that way?" or "What makes your plan better than mine?" In some way the person would have to convey either their planning objective $\beta$ or their plan route $P_H$ to the robot so that VBMO could interpret it in its shared graph and evaluate it with respect to all the objectives in $B$. The same process would then compare VBMO's selected plan to $P_H$ with respect to $B$. Future work could incorporate these explanations in a complete dialogue system for natural conversation with person (e.g., (Krarup et al. 2021)).

Both evolutionary methods and VBMO consider a population of solutions and select among them with a kind of fitness function. VBMO, however, does not require multiple iterations to refine its plan. Instead, it starts with at least one plan already on the Pareto frontier and uses a shared underlying graph to select a plan that is generally expected to perform well with respect to all the objectives. Although VBMO does not need hand-tuned weights to balance multi-

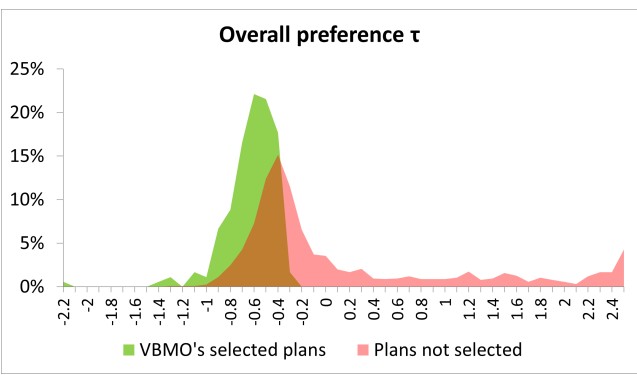

Figure 5: Distribution of overall preference $\tau$ for 200 selected plans. $\tau$ is a *t*-statistic and so has mean 0. A selected plan must have minimum total score, so its $\tau$ value must be negative unless all plans score the same on every objective.

Table 4: How confidence metric sentiments co-occur.

| $\mathcal{L}(\tau)$ | All plans $\mathcal{L}(\alpha)$ | | | Total |
|---|---|---|---|---|
| | certain | somewhat certain | conflicted | |
| really | 3.5% | 3.2% | 1.5% | 8.2% |
| somewhat | 11.0% | 13.7% | 1.7% | 26.4% |
| not really | 20.4% | 28.1% | 17.0% | 65.5% |
| | $P_{best}$ | | | Total |
| | certain | somewhat certain | conflicted | |
| really | 13.3% | 12.2% | 3.9% | 29.4% |
| somewhat | 24.9% | 25.4% | 1.1% | 51.4% |
| not really | 12.7% | 6.6% | 0.0% | 19.3% |

ple objectives, they could be easily incorporated into equation 1 to change an objective's influence on the sum.

Currently the planners modify a graph with respect to features of a learned model and commonsense rationales, but additional planning objectives could be incorporated to produce more nuanced explanations and create more robust plans. For example, path *smoothness* (how well a path maintains a straight trajectory) is an important criterion for indoor navigation, particularly for transport of fragile material. Smoothness could be readily translated into a planning objective, so that it explicitly impacts plan selection. VBMO uses A* for its graph search algorithm but another optimal graph search algorithm could easily be substituted for it.

Because the controller's learned spatial model becomes more knowledgeable as the robot experiences the environment, the planners that rely on the model become better informed over time. As a result, early in an experiment's 40 tasks, planners that reference the spatial model produce the same plan as FAST and so their objectives are often excluded from explanations because they do not differentiate between the two plans. As a consequence, the average readability of explanations for the first 5 targets is 6.8 but increases linearly to 8.0 for the last 5 targets. The current explanation process is efficient; it is linear in the number of objectives. With the 8 objectives here, for example, average response time for the two questions together was less than 10 msec.

In previous work we generated contrastive explanations but assumed the alternative objective was fixed, and only compared performance on the selected plan's objective and the fixed objective (Korpan and Epstein 2018). That approach produced shorter, more understandable explanations but did not address the nuances of the multiple objectives under consideration by VBMO. Here, the explanations averaged 51.2 words, which may be too long to hold a person's attention or limit their understanding. One way to shorten VBMO's explanations would be to include only objectives that strongly favor or oppose a plan, so that only the most important reasons are highlighted. Although some objectives' plans may never be selected by VBMO (e.g., NOVEL), their rationales may still be significant for selection among more desirable plans. This is similar to the way people consider multiple reasons during decision making and weigh their importance differently (Judd and Lusk 1984; Wilson and Schooler 1991).

Current work evaluates our contrastive explanations with human subjects to gauge how well they are trusted and understood. This study evaluates the language associated with VBMO's planning objectives. It will also test the hypothesis that constrastive explanations improve trust and understanding more in comparison with single-objective explanations. Future work could identify an upper bound to the number of objectives used in a plan explanation before understanding deteriorates.

VBMO can encounter a task where its set of objectives $B$ generate plans that are equally poor on all the other objectives. In that case, total scores for all plans are equal and VBMO selects a plan at random, one that should perform well only on its own objective and poorly on the others. That reduces the solution to a planner with a single, randomly chosen objective. Other multi-objective planning methods avoid this difficulty by compromise among all the objectives rather than focus on strong performance from one. To address this issue, VBMO could incorporate additional planners that introduce weighted sums of different objectives so that the planner is forced to find a plan that compromises between them but would do less well on any single objective. Unless these additions were simple, they would make it more difficult to generate natural language that explains those weighted sums.

Although VBMO is applied here to path planning for robot navigation, it is more generally applicable to any multi-objective planning problem (e.g., motion or task planning). The contrastive explanation approach described here could also apply to many multi-objective domains; one would first generate an optimal single-objective solution for each objective, and then evaluate the resultant solutions with each of the other objectives. The same metrics used here would then apply. For example, a movie recommendation system could identify the movie most similar to a user's top favorites, the most popular movie, and the best reviewed movie, evaluate each of them on the other objectives, and then select the movie that scores best across the three. A

contrastive explanation that might be generated is "Although another movie may be much more popular, I recommend this one because it is somewhat better reviewed and a lot more similar to your favorite movies."

Meanwhile, VBMO is an inherently interpretable approach that generates plans on the Pareto frontier. The plan it selects is guaranteed to be optimal with respect to at least one objective and likely does well on the others. VBMO easily generates contrastive explanations in natural language. These explanations flexibly compare plans with respect to the objectives under consideration and express the controller's confidence in its selected plan. The results with voting-based multi-objective path planning presented here demonstrate that explainable AI planning algorithms need not sacrifice optimality to be well understood.

## Acknowledgements

We thank the anonymous reviewers and the conference participants for their constructive and insightful comments and suggestions.

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
