# OpenReview forum: "Contrastive Natural Language Explanations for Multi-Objective Path Planning"
_icaps-conference.org/ICAPS/2021/Workshop/XAIP — XAIP 2021_

### Official Review · AnonReviewer1 · 2021-06-26
**It was a little challenging to read this paper. Although some of the claims are a little questionable, this can turn into a conference paper with some work, so I suggest including it in the workshop.**

**Rating:** 6
**Confidence:** 4

**Review:**

**Summary:** This paper presents a unique approach to generate contrastive explanations for navigation plans based on multiple objectives. The authors validate their approach within the simulation to produce diverse and readily understandable explanations.

**Strength:**

(+) Technical stack and implementation details are very well laid. I really like the tables the authors provided. I would suggest authors reference them more for clarity while explaining their approach.

(+) The contribution and significance of the work is relevant to explainable AI


**Suggestions and Questions to the Authors:**

1. I am a little concerned with some of the claims which abstract and authors assert. Specifically, in the introduction, the authors introduce the approach as “flexible, scalable.” Still, I do not see any theoretical or empirical evaluation that studies how this approach scales with the number of objectives, plans, etc.  My question and suggestion to authors would be to address the scalability and runtime concerns regarding: 1) generation of different plans as they increase, and 2) increasing objectives.
2. Another claim is testing for the understandability of explanations within the abstract. I do not see any human subject study which tests the usability or understandability of explanations. I agree that the explanations presented in the current work are human-readable/understandable. Still, we cannot ascertain usability/understandability claims within the application domain without explicitly testing them on users.
3. Furthermore, I would like to question the authors on how comprehensive these explanations would be as the objectives in plans increase. For example, one of the plan given in the paper -- "Although another way may be a lot more well-traveled, a lot closer to ways we've gone before, somewhat more aligned with previous routes, and somewhat less familiar, I believe my way is a little more likely to reveal new areas, a lot more open, a lot shorter, and a lot less obstructed." This example is slowly reaching a level where it is little taxing for the user to understand the reasoning\motivation behind the agent's decision. Do authors think there might be some upper bound after which users cannot keep track of different objectives within the plan explanations?
4. In the related work, authors claim explanations based on state-action-reward representation are not human-friendly, which is an overstatement. I would recommend looking into work from Hayes and Shah (2017) and more.
- Hayes, Bradley, and Julie A. Shah. "Improving robot controller transparency through autonomous policy explanation." 2017 12th ACM/IEEE International Conference on Human-Robot Interaction (HRI. IEEE, 2017.
- Wallkotter, S., Tulli, S., Castellano, G., Paiva, A., & Chetouani, M. (2020). Explainable agents through social cues: A review. arXiv preprint arXiv:2003.05251.
- Tabrez, Aaquib, Shivendra Agrawal, and Bradley Hayes. "Explanation-based reward coaching to improve human performance via reinforcement learning." 2019 14th ACM/IEEE International Conference on Human-Robot Interaction (HRI). IEEE, 2019.
5. I would like to know the authors' thoughts on how single objective explanations would fare with multi-objective explanations in real-world usage. Explanations provided by authors here are essentially justifications\rationale. Justifications do not necessarily aim to explain the actual decision-making process (Ehsan et al. 2019). People might be more accepting of systems that can give simpler justifications compared to complicated multi-objective explanations—extrapolating from Tim Millers' seminal review, "Explanation in artificial intelligence: Insights from the social sciences. Artificial intelligence": People expect not only contrastive explanations but also need selective causal explanations.

---

### Official Review · AnonReviewer2 · 2021-07-06
**Interesting paper, important topic but maybe a bit premature and could be better positioned.**

**Rating:** 6
**Confidence:** 3

**Review:**

Explainable planning is a vital field of study and, while planners are often held up as explainable (in contrast to most statistical machine learning techniques such as deep learning), there have not been as many papers, like this, that discuss how to actually extract useful explanations from planning systems.

In my view, while I am very keen on the topic of this research and I think that this research in general should definitely be published, I think that there are two major aspects of this work, that are not covered in this paper, that are necessary for a strong publication.

First, the evaluation of the work is lacking. There are, for sure, many metrics and graphs shown (albeit without error bars). However, the most important evaluation, being if the explanation is actually useful for a particular purpose, is left to future work. In my view, without at least some discussion about who might find this useful and why, perhaps with even a small set of data, this paper becomes quite weak. Unfortunately these evaluations are very hard and inconvenient to perform because they usually involve actually asking people to evaluate these explanations, but such is the cost of working in a field that, by necessity, crosses into human factors.

I would be less worried about the lack of such evaluations if there was, instead, a better positioning of the explanations in terms of who would actually find them useful. Explanations are targeted to a particular audience; the explanations that are useful to an engineer, an end user, or a forensic investigator all differ. The authors have developed a way of generating contrastive explanations from the planner and contend that they are better than "other explanations" but haven't really explained what the human is actually using the explanation for. Is it to make the human feel better about trusting the system? To help the human better predict what the system will do next time? To teach the human something? To help the human figure out how to stop the system from making a mistake next time? Without knowing this, it isn't clear that the developed system could be properly evaluated, or that the central assertion that contrastive explanations are more desirable than "other explanations" is well founded.


I have a few additional concerns.

I'm a bit surprised that on page 4, it is contended that the case for building interpretable models has been "recently argued". Building AI models that are intrinsically explainable has been a topic of study that goes back decades. Expert systems, inductive logic programming, and behavioral cloning are but examples of areas of AI with long histories where there has been a focus on the building of explainable models. Getting explanations out of black boxes is a much more recent phenomena, made more topical by the rapid advancement in statistical machine learning systems in general, and deep learning in particular.

Voting based ensemble methods, such as VBMO, are, by definition, dominated by the components (planner objectives, in this case) that have some form of agreement, and this is evident in their discussion - for instance, "NOVEL" is never chosen because it generally conflicts with the majority. This is pointed out, but oddly enough not in the context of why you would then bother with it but rather, in the context of a bunch of statistics that, while interesting, aren't actually that useful in determining if the resulting explanations are useful. I think there is actually a very good reason to have it in there, which is for its influence on the explanation generation, but that would require a discussion as to how the explanations are actually useful for a particular purpose, which doesn't really appear in this paper.

The reported statistics are also rather odd. For instance, I'm not sure why "number of words" is, in isolation, an interesting statistic. A shorter explanation might hold the user's attention more, but of course perhaps they would rather a longer explanation that is useful, rather than a shorter but less useful one. However, the match between the underlying information and the text of the explanation is pretty arbitrary (because it depends on the phrasing that the implementers have chosen) and there is no real discussion that allows me to figure out if the result is good or bad. There is, in my view, similarly insufficient explanation as to why the other statistics are meaningful *and* useful.

---

### Meta-Review · Area_Chairs · 2021-07-07

**Recommendation:** Accept
**Confidence:** 3

**Metareview:**

Thanks very much for submitting your paper!

Summary: The paper tackled the problem of generating contrasting multi-objective explanations for path planning by comparing single-objective plans.

Strengths:
- Relevant topic
- Detailed implementation
- Convincing methodology

Limitations:
- Missing human subject study to test understandability of the generated explanations
- Poor information about scalability and runtime
- Limited information about how multi-objective explanations would apply in real-world settings. This aspect is also linked with an unclear target audience for the explanations.

We hope that you find the reviewers' comments to be informative, please take them into account when revising your papers. We look forward to your presentation!

---

### Decision · Program_Chairs · 2021-07-08

Accept